# Non-Invasive Cardiac Output Monitoring with Electrical Cardiometry During Laparoscopic Cholecystectomy Surgery, a Cross-Sectional Study

**DOI:** 10.3390/jcm14072228

**Published:** 2025-03-25

**Authors:** Khaled Ahmed Yassen, Walla Aljumaiy, Imran Alherz, Lina A. AlMudayris, Sara Abdulhameed AlBunyan, Renad S. AlSubaie, Fatma Alniniya, Sherif Saleh

**Affiliations:** 1Anesthesia Unit, Surgery Department, College of Medicine, King Faisal University, Alahsa 31982, Saudi Arabia; 2Anaesthesia Department, King Fahad Hospital, Hofuf 36441, Saudi Arabia; dr.w.hussain@hotmail.com (W.A.); majbourha@gmail.com (F.A.); 3AlAhsa Health Cluster, Anaesthesia Department, King Fahad Hospital, Hofuf 36441, Saudi Arabia; dr.imran.alherz@gmail.com; 4Alumni College of Medicine, King Faisal University, Alahsa 31982, Saudi Arabia; leenaalmudaires@gmail.com (L.A.A.); susuhmb@gmail.com (S.A.A.); 5College of Medicine, King Faisal University, Alahsa 31982, Saudi Arabia; subaierenad@gmail.com; 6Surgery Department, College of Medicine, King Faisal University, Alahsa 31982, Saudi Arabia; ssaleh@kfu.edu.sa

**Keywords:** hemodynamics, cardiac output, laparoscopic cholecystectomy, electric cardiometry

## Abstract

**Background**: Increased intra-abdominal pressure (IAP), autonomic reactions, and anesthetics all contribute to hemodynamic alterations during laparoscopic cholecystectomy. This study’s objectives are to measure noninvasively the intraoperative individual responses in cardiac and systemic hemodynamics, focusing on cardiac output (CO. L/min), stroke volume (SV, mL/min), systemic vascular resistance (SVR, dyn.s.cm^−5^), and noninvasive mean arterial blood pressure (MABP, mmHg) during and after peritoneal insufflation (cmH_2_O). The secondary objective was to evaluate the utility of EC as an adjunct to standard monitoring and to assess the individual differences. **Methods**: The CO and associated parameters were continuously and noninvasively monitored with the electrical cardiometry (EC, ICON, Osypka, Berlin Germany). **Results**: Seventy-three patients showed that when the IAP increased to 13 [IQR: 13-14] cmH_2_O, there was an overall percentage decrease in CO (−11.29%), MABP (−9.31%), and SVR (−23.16%) compared to pre induction with minimal changes in heart rate (HR). Individual variation and extreme reactions among certain patients were noted, with CO falling by −47.14% and MABP by −61.59, respectively, which can have major repercussions. **Conclusions**: The EC enabled real-time, non-invasive CO monitoring and detected significant cardio-hemodynamic changes that conventional monitors could miss. EC can supplement traditional monitors and give attending anesthesiologists access to more of patients’ vital information.

## 1. Introduction

Numerous factors may affect the hemodynamics during laparoscopic cholecystectomy.

The increase in intra-abdominal pressure (IAP) reduces the venous return (preload), which affects the cardiac output (CO). The sympathetic nervous system activation in response to the surgical stress (catecholamine release) increases heart rate and systemic vascular resistance (SVR) to counteract the drop in CO and to maintain blood pressure. The reverse Trendelenburg position (head-up) and the changes in autonomic tone can predispose to arrhythmia, and not least to mention, the general anesthetics decrease the SVR and depress the CO. All these multi-factors will likely vary from patient to patient [1,2,3,4]. Significant cardiovascular changes may go unnoticed if not adequately monitored [5,6]. There have been multiple reports of abrupt intraoperative cardiovascular events during a smooth laparoscopic cholecystectomy, as well as documented unfavorable cardiovascular, hormonal, and neuroendocrine changes, brought on by elevated IAP [7,8]. Traditional systemic monitors cannot measure cardiac functions or identify significant dysfunctions, and invasive catheter placement for cardiac function monitoring during routine laparoscopic cholecystectomy cannot be justified unless there is significant cardiovascular disease dysfunction prior to surgery.

Now it is possible to monitor a continuous, beat-by-beat CO by the electrical cardiometry (EC) (ICON monitor; Cardiotronics Inc., La Jolla, CA, USA), which is a noninvasive cardiac output monitor that identifies changes in thoracic electrical bioimpedance brought on by changes in blood conductivity of the aorta [9,10]. Heart rate and blood pressure do not provide a full picture of a patient’s hemodynamics. The gaps in traditional monitoring can be filled by the parameters that EC offers. Fluid resuscitation and medication therapy can be guided in a targeted, continuous way by EC parameters such CO and SV measurements, which offer additional information on preload, contractility, after-load, and delivered oxygen. The “gold standard” techniques like thermodilution have been used to validate electrical cardiometry [9]. The primary aim of this study is to examine the intraoperative cardiac output fluctuations and systemic hemodynamics that associate peritoneal insufflation and increased IAP during and after laparoscopic cholecystectomy with both EC and standard monitors. The secondary aim is to evaluate the utility of EC as an adjunct to standard monitoring and to assess the individual differences.

## 2. Methods

After receiving approval from the local ethics and research committee Alahsa Health Cluster (IRB Log No.:15-EP-2024) on 21 September 2024, a non-experimental cross-sectional study was carried out at King Fahad Hospital in Hofuf, Saudi Arabia. All patients signed an informed consent form prior to participation. The trial took place from 2 October 2024, to 29 January 2025. Consecutive adult patients with ASA Physical Status Classification I–II (age > 18 year) and scheduled for an elective laparoscopic cholecystectomy were included. Patients with cardiovascular or hemodynamic instability before surgery were not included. Any serious cardiovascular disease or cardiac arrhythmias may interfere with EC signals, which is the medical basis for the exclusions listed above. Additional exclusion criteria were patients with significant coagulation problems, those on oral anticoagulants or antithrombotic medications, those with a BMI of 40 or higher, and those who had experienced intraoperative complications such as conversion to open surgery, air embolism, or serious bleeding.

In the outpatient Anesthesia Clinic, the patients were assessed for the procedure and treated in accordance with the established and approved criteria. No decisions about diagnosis or treatment were made using the data from the EC device because the treating physicians were blinded to the information gathered.

### 2.1. Sample Size Calculation

The minimum sample size was determined based on a previous study which aimed to prospectively examine the effects of pneumoperitoneum on CO and hemodynamics during laparoscopic surgery [11]. Banerjee et al. in 2021 concluded that patients undergoing laparoscopic cholecystectomy experience hemodynamic changes after pneumoperitoneum insufflation and increased IAP. The sample size was calculated to detect the EC as a non-invasive monitor for CO, and for systemic hemodynamics during laparoscopic surgery. At a power of study of 80% (b error accepted = 0.20), and difference in proportion of 10%, a minimal required sample size was found to be 57 patients [12].

### 2.2. Electric Cardiometry (EC)

EC is an algorithm for estimating stroke volume (SV) and CO from the non-invasive and continuous measurements of thoracic bioimpedance [11,13,14]. It specifically detects fluctuations in thoracic electrical bioimpedance resulting from variations in blood conductivity within the aorta, attributed to the dynamic reorientation of red blood cells (RBCs) during the cardiac cycle (Figure 1) [4,5,9,10]. The monitor requires the placement of four skin sensors on the neck and left side of the thorax which allow for continuous measurement of changes of electrical conductivity throsugh the thorax after applying low amplitude, high frequency electrical current [15,16]. EC is one of the new modalities to evaluate the fluid status and responsiveness. Systolic pressure variation (difference between maximum and minimum systolic pressure during one mechanical breath), known as stroke volume variation (SVV), can highly predict fluid responsiveness [11,17].

### 2.3. Anesthetic Technique

General anesthesia was induced with propofol 1.5–2 mg/kg, rocuronium 0.6 mg/kg, and fentanyl 1–2 μg/kg intravenously, following pre-oxygenation with an O_2_/Air mixture (FiO_2_ = 0.8) and inhaled sevoflurane to keep the anesthesia depth between 25 and 50 as per SEDLine anesthesia depth monitor (Masimo, Irvine, CA, USA), and boluses of rocuronium injections of 0.1–0.15 mg/kg were administered every 30 min to keep muscles relaxed. To ensure proper oxygenation (SaO_2_, > 96%) and maintain the end tidal carbon dioxide between 35 and 40 mmHg, the following ventilation parameters were maintained following endotracheal intubation: tidal volume (Vt) 8 mL/kg, Positive End-Expiratory Pressure (PEEP) 5 cmH_2_O, a respiratory rate of 12 breaths/min, and peak pressure (P peak) kept below 40 mmHg.

After receiving carbon dioxide insufflation through the peritoneum, the patient was placed in a head-up tilt position of 30 degrees. Ringer acetate was infused intraoperatively at rate (0–10 mL/kg/h) to keep Pleth. Variability Index (PVI %) (Masimo, Irvine, CA, USA) <12%, and if PVI were >12% despite crystalloids infusion, then boluses of 3 mL/kg colloid solution (Albumin 5%) were administered. Standard anesthesia monitoring included electrocardiography, noninvasive blood pressure, pulse oximetry, ventilation parameters, capnography, FiO_2_ (0.5), and IAP (<14 cmH_2_O). Any decrease in blood pressure was controlled by lowering the level of anesthesia and maintaining the PVI first, followed by the administration of catecholamine support if required. Reducing and even ceasing any increase in IAP or the insufflation process is always available for patient safety, if severe drops in hemodynamics were reported.

### 2.4. Retrieved Data and Measurement Times

Biographical data: Age (years), gender (M/F), presence of co-existing diseases, and the American Society of Anesthesiologist classification (ASA) for each patient were reported. Hemodynamic parameters: Heart rate (beat/min) and MABP (mmHg). EC parameters: CO (L/min) (CO), SV (mL), systemic vascular resistance (dyn.s.cm^−5^) (SVR), and stroke volume variation (SVV, %). Other parameters: IAP in cmH_2_O, total intraoperative volume of crystalloid and colloids consumed (mL), hemodynamic instability, oxygen desaturation, postoperative nausea and vomiting, and catecholamine consumption were all documented.

Measurement times: T0: Pre- induction; T1: post intubation; T2: post insufflation and positioning; T3: 15 min post insufflation; T4: 30 min post insufflation; T5: end of surgery; T6: 10 min post deflation; T7: post extubation.

### 2.5. Statistical Methodology

Both manual and automated charting techniques were used to record data, and digital files were used to store operational information. The study team verified and documented all of the data that were gathered. Data were collected and entered into the computer using the SPSS 25 (Statistical Package for Social Science) program for statistical analysis (ver 25) [18]. Most of the continuous variables were not-normally distributed (Shapiro–Wilk test) [19], so the non-parametric statistics was adopted [20]. Data were described using minimum, maximum, mean, standard deviation, median, 95% CI of the median, and 25th to 75th percentiles (inter-quartile range (IQR)). Friedman’s test was used for intra-group comparisons [21]. Non-parametric Kendall’s tau correlation (τ) was used [19]. Intra-class correlation (ICC) was used to assess agreement [22,23]. During sample size calculation, beta error accepted up to 20% with a power of study of 80%. An alpha level was set to 5% with a significance level of 95%. Statistical significance was tested at *p* value < 0.05 [24,25].

## 3. Results

Seventy-six (n = 76) patients were initially enrolled, but only seventy-three (n = 73) were included and finished the study. Defective electrodes conduction were the reason for the three exclusions. The demographics of the 73 included patients were 53 (72.6%) females and 20 (27.4%) males, with a mean ± age of 42.6 ± 9.8 years, body mass index (BMI) averaging 26.9 [23.98–32.14] kg/m^2^, 28.8% had hypertension, 27.4% were men, and 38.4% were ASA I.

### 3.1. Main Findings

One of the main finding was the depression of the CO and MABP with anesthesia induction and peritoneal insufflation when compared to the baseline (T0) and the immediate recovery with deflation (Figure 2 and Figure 3). Individual differences were visible in the results, with extreme reactions (negative percentage depression) (Table 1 and Table 2). In specific patients, the CO decreased by −47.14%, and MABP by −61.59% post insufflation at 15 and 30 min, respectively (T3 and T4), compared to T0 baseline (Table 2).

Over the course of the procedure, eight patients experienced a 40% or more drop in CO. Three were males, and five were females with co-existing diseases. Five (62.0%) of the eight had a comorbid illness (three with hypertension, and four obese).

This significant drop in MABP (anesthesiologists blinded to EC CO) was managed by reducing the anesthesia depth until the MABP recovered without the need for any catecholamine support or extra fluids in any patients. The mean ± SD of the IAP was at 13.26 ± 0.50 cmH_2_O (median interquartile 13 [13-14] cmH_2_O) with carbon-dioxide gas peritoneal insufflation, and this was maintained during surgery. The volume of crystalloids infused during surgery was 850 ± 120 mL guided by the PVI% to maintain adequate normovolemia, with 26.0% of the patients requiring colloid boluses. No major complications were reported, and recovery was uneventful. No paradoxical hypertensive responses to insufflation were reported. Postoperative nausea and vomiting was only reported in two (2.74%) of the studied patients.

Percentage change was calculated as follows:$$Percentage\;change\;\left(\%\right)=\frac{Measurement\;\left(any\;T\right)-Meaurement\;\left(T0\right)}{Meaurement\;\left(T0\right)}\times 100$$

### 3.2. Hypertensive vs. Normotensive Patients

Both SV and CO trend variations over time were not impacted by inflation or deflation in HTN patients (*p* = 0.8, *p* = 0.9, respectively); unlike patients with no HTN (*p* < 0.01), this could be attributed to the preserved vascular tone among HTN patients, Figure 4.

At measured points, hypertensive (HTN) patients (28.8% of the studied patients) had a greater SVR than non-hypertensive; however, this difference did not achieve statistical significance, with the exception of baseline (T0) (*p* = 0.004) and following peritoneal insufflation T2 (*p* = 0.02), Figure 5. The greater vascular tone that hypertension patients already have prior to surgery and following insufflation may be the cause of the elevated SVR. Before surgery, all ASA II hypertension patients were under control with pharmacological agents (beta-blockers and angiotensin receptor blockers, arbs) and instructed to take their morning antihypertensive dose.

The limited sample size of the HTN and normotensive subgroups in the current study may be the cause of these mathematical variations observed between them in CO, SV, and SVR and, which did not approach statistical significance. A post hoc subgroup analysis was conducted, when these differences were observed, and suggested that a large number of patients was necessary to confirm these findings.

### 3.3. Stroke Volume Variation and Pleth Variability Index

Figure 6 shows a correlation and agreement between the stroke volume variation (SVV, %) of EC and the Pleth Variability Index (PVI, %, Masimo, Irvine, CA, USA), but the clinical association was poor despite the statistical significance, based on the rule of thumb for interpreting the size of a correlation coefficient by Hinkle DE et al. (2003) [26].

There is very little bias (mean difference = 0) between the 584 measurements in this study; the precision (precision) limits are wide (−14.6 to 14.5). The agreement lacks any particular pattern and is uniform. A low positive agreement was found between the SVV and PVI measurements (ICC = 0.367, 95% CI: 0.225–0.46).

## 4. Discussion

This study emphasized the hemodynamic changes that took place during laparoscopic cholecystectomy, and by the elevated IAP. The results showed that following anesthesia induction and during peritoneal insufflation, the CO and MABP were reduced. These results are consistent with that of other studies such as Wittgen et al. (1991), Dexter et al. (1999), Jin et al. (2021), and Gannedahl et al. (1996) [12,27,28,29,30]. EC succeeded in offering a real-time, non-invasive way to track cardiac functions and highlighted individual differences in hemodynamic responses that conventional monitors could have overlooked. EC’s beat-by-beat surveillance of the patient’s cardiac and systemic hemodynamic changes adds value to conventional monitors by enabling early detection of any emerging abnormalities in the cardiovascular system function. EC can detect the hemodynamic changes on the spot, while other routine monitoring provides delayed results when the body fails to adjust. In contrast to EC, medications that alter vascular tone, such as vasopressors, can drastically affect the performance of pulse oximetry and PVI, which are routinely used among the standard monitors. Patients with severe hypovolemia and inadequate peripheral perfusion respond best to EC, whereas PVI is unreliable in these conditions.

The physiological knowledge that elevated IAP compresses the inferior vena cava and lowers the venous return (preload) is in line with that reported by Łagosz P et al., in 2022 [31]. According to Łagosz P et al., the increase in IAP ultimately results in a decrease in CO and SV. The results from our current study indicated that pneumoperitoneum and general anesthesia led to individual and notable decrease in CO (−47.14%) and MABP (−61.59%) in specific patients as evident in Table 2. These responses are especially important for high-risk cardiac patients because such hemodynamic instability can lead to unfavorable cardiovascular outcomes, such as myocardial ischemia or arrhythmias. Atkinson et al. (2017) in their study came to the conclusion that the hemodynamic effects are increased in patients with cardiovascular disease such as ischemic heart disease, valvular, and congenital heart disease [2].

The results analysis also identified subgroups of patients with and without hypertension in the current study.

Hemodynamic changes are known to be more pronounced in hypertensive patients than normotensive patients. However, in the current study, the CO trend changes in HTN patients were not substantially affected by inflation or deflation (*p* = 0.8). This may be because the HTN patients respond differently due to their altered vascular elasticity. Our inclusion criteria only include ASA I–II patients with controlled hypertension (ASA II). Patients included were on beta-blockers and angiotensin receptor blockers (arbs) therapy. The blunted baroreceptor response, as a result of these medications, could have helped to prevent significant fluctuations in CO and MABP during peritoneal insufflation. Kim et al. (2010) came to the same conclusion when they studied hypertensive patients during laparoscopic colectomy, and their results indicated that pneumoperitoneum does not lead to clinically negative hemodynamic changes in heart rate, mean arterial pressure, or cardiac output of hypertensive patients who have taken antihypertensive drugs for more than 1 month [32]. Interestingly, the hypertensive patients in our study (all ASA II) were on daily beta-blockers and arbs prior to the scheduled surgery and instructed to take their morning doses. They showed minimal changes in CO and MABP, unlike normotensive patients.

These findings suggest that well-controlled hypertensive patients (ASA I–II) may not be at significantly increased perioperative risk. However, it is essential to emphasize that further studies are needed to determine whether this trend persists in ASA III-IV patients with severe hypertension or with other cardiovascular comorbidities.

No documented hypertensive episodes in the current study were reported; it is challenging to assess the impact of these drugs (beta-blockers and arbs) on the patients’ hemodynamic response based on the number of patients in the current study.

To validate these findings in our research, a larger-scale investigation is advised as confirmed by the post hoc analysis. In support of the above, recent research by Zhang et al. in 2023 [33] also suggested that hypertensive patients may react differently to high IAP, and they attributed this to their low vascular elasticity and adaptation. The potential for more personalized intraoperative care in patients with co-existing diseases such as hypertension should be encouraged.

This current study’s use of EC marks an improvement in perioperative monitoring. The utilization of invasive arterial lines or pulmonary artery catheters, to identify minute alterations in cardiac function, especially in real-time, cannot be justified for this type of surgery (laparoscopic cholecystectomy). Contrarily, EC offers an alternative non-invasive CO monitoring, empowering anesthesiologists to make better choices during surgery with minimal risk to the patients. Given the potential for sudden hemodynamic changes during laparoscopic surgeries, the current study found significant individual diversity in hemodynamic responses to high IAP, notwithstanding the general patterns. Extreme reactions were seen in specific patients, with CO and MABP falling by more than 40% as demonstrated in Table 2. These results highlight how crucial customized hemodynamic care is during laparoscopic procedures, especially for patients who already have cardiac co-existing diseases. In 2021, Banerjee et al. evaluated the hemodynamic changes during laparoscopic cholecystectomy by transthoracic echocardiography. They reported a fall in CO by 45%, *p* < 0.001, and SV by 42%, *p* < 0.001, with pneumoperitoneum, as well a significant rise in MAP (11%, *p* < 0.001). In the current study, the overall drop in CO was −11.29%, and in specific patients the depression of CO could reach −47.14% [12]. The MABP in our current study found a −9.31% decrease and was associated with reduced SVR. Physiologically, the increase in IAP compresses the inferior vena cava and the renal vessels which releases vasopressin, adrenaline, noradrenaline, and atrial natriuretic peptide. These hormones increase peripheral vascular resistance and blood pressure, thus increasing left ventricular afterload and the heart rate to compensate. The failure of the heart rate (HR) and SVR in the current study to increase in response to the increase in IAP could explain this reduction in MABP observed in our results (Table 1 and Table 2).

The failure of SVR to increase could be due to IAP, which was not extremely elevated to above 15 mmHg, as the mean (SD) increase in IAP was only 13.26 ± 0.50 cmH_2_O.

Lee-Ong, A. (2023) stated in his study that IAPs above 15 mmHg decrease significantly the venous return to the inferior vena cava and compress the renal vessels, which lead to the reduction in renal blood flow and the release of vasopressin, adrenaline, noradrenaline, and atrial natriuretic peptide. These hormones increase peripheral vascular resilience and blood pressure, thus increasing SVR. In our current study the IAP was lower than 15 mmHg, and this could explain why SVR was not elevated [34].

Hirvonen et al. [35] detected a 20% reduction in CO during pneumoperitoneum with transesophageal Doppler imaging, and Alishahi et al. [36] reported a similar reduction in CO with pneumoperitoneum. Russo found that pneumoperitoneum has important effects on left ventricular volumes, causing a drop in left ventricular end-diastolic volume [37].

In contrast, Larsen et al. found that carbon dioxide pneumoperitoneum insufflation increased preload and afterload in patients undergoing laparoscopic cholecystectomy surgery, which decreased heart performance (fractional shortening), but there was no significant drop in CO from pneumoperitoneum [3]. This could be explained by the compensatory and significant increase in HR. In our current study, the heart rate was not increased but instead was clinically stable as evident from results (Table 1), and this could have contributed to the reduction in CO and MABP.

In reaction to the decreased venous return brought on by the elevated IAP, the increase in SVR is probably a compensatory mechanism to preserve perfusion pressure as explained by Joris et al. [38]. This was not the case in our current study, as the drop in CO and SV was not associated with a compensatory increase in SVR or HR, but instead the reduced venous return with increased IAP could only be blamed for the reduction in CO. The current study results are in line with research by Gannedahl et al. [30], Odeberg et al. [4], and Biswas et al.’s study (2020) [39].

### 4.1. EC Precision

The results of the current study demonstrated that EC was successful in measuring the trend changes in CO throughout time. However, in some studies, the absolute values of the CO measured using the EC and the CO measured using the thermodilution approach were interchangeable, but in other studies, they were not.

The variations in clinical situations have an impact on EC CO’s performance. In some therapeutic scenarios, such as during patent ductus arteriousus (PDA) ligation in operating rooms and during transportations, EC CO trend changes were very helpful.

Nevertheless, a number of studies examining the accuracy of the absolute EC CO yielded conflicting results. A recent meta-analysis by Sanders et al. in 2020 found a wide range of bias with EC absolute CO values [40], but one of its limitations was that it included both adults and children. In 2015, Suehiro et al. [41] carried out a systematic review and meta-analysis that looked at the accuracy of several minimally invasive CO devices, including EC. They discovered that EC had the lowest bias and the lowest mistake percentage, especially in cardiac surgery.

Three years later in 2018, Altamirano-Diaz et al. demonstrated that the CO measurements by EC in pediatric patients undergoing coarctation of aorta repair were equivalent to those by the transoesophageal echocardiography (TEE) provided that no increase in left ventricular output (LVO) is present. [42]. While it is not appropriate to directly replace the gold standard cardiac output monitoring techniques (thermodilution), this EC noninvasive and readily adaptable method may offer valuable dynamic information on hemodynamic trend events during anesthesia.

### 4.2. Limitations

This study has several limitations. The sample size, particularly in the subgroup analysis, was insufficient to detect statistically significant differences between hypertensive and normotensive patients. Additionally, the study only examined laparoscopic cholecystectomy and no other laparoscopic surgeries. The selective inclusion of patients classified as ASA I-II may restrict the applicability of the device in higher-risk patients (ASA III and above), which should be investigated in any further studies.

The reliability of the measured values need careful interpretation, because no second CO measurement method was used. However, the trending changes demonstrated in results were more informative than absolute values and were compatible with the meta-analysis by Sanders and colleagues indicating that EC could be an addition to the standard monitoring by providing a continuous and specific cardiac function trend changes which other monitors lack [4]. Few technical challenges were noted during the study, such as electrodes misconductance and/or electric diathermy signal interference. The electrocautery diathermy utilized during surgery interfered with the EC signals. Cardiac arrhythmia also affected the signaling to the EC electrodes. Further research is needed to address these limitations.

The additional cost of cardiac output monitoring can be justified for patients with cardiac risk or patients undergoing major operations, but the cost for routine operations like laparoscopic cholecystectomy needs to be addressed and lowered.

One single-use disposable EC skin electrodes set costs 650 Saudi Arabia Riyals (SAR) for each patient in the Kingdom of Saudi Arabia (173.3 US Dollar), whereas the device monitor costs 90,000 SAR (or Free for every 400 disposable set). The near-future registration of the product with the National Unified Procurement Company (NUPCO), an organization owned by the Saudi Government Public Investment Fund (https://nupco.com/en/) accessed on 11 February 2025, will reduce the costs further. NUPCO as an organization is expressly tasked with determining and reducing the costs of healthcare services throughout Saudi Arabian hospitals.

## 5. Conclusions

This study shows how elevated IAP during laparoscopic cholecystectomy has an effect on cardiac and systemic hemodynamics, mainly decreasing cardiac output and mean arterial blood pressure. EC detected significant hemodynamic trend changes that conventional monitors could miss. EC demonstrated its potential as an additional useful tool in perioperative monitoring. Its non-invasive nature, combined with real-time cardiac and hemodynamic monitoring, makes it an attractive option for ensuring patient safety beside standard monitors.

The individual hemodynamic differences in response to the increase in IAP highlight the necessity of tailored hemodynamic management strategies for each patient. Additional investigation is necessary to examine these results in broader, more varied laparoscopic surgeries and patient groups particularly to look in future at the consequences for patients with high-cardiac risk and severe hypertension.

## Figures and Tables

**Figure 1 jcm-14-02228-f001:**
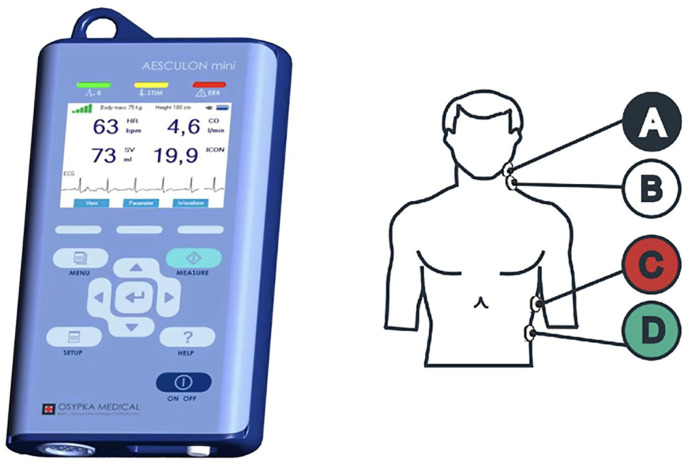
Electric cardiometry monitor and electrodes anatomical placements. Electrical cardiometry (ICON Osypka, Germany) provided continuous, non-invasive measurements of cardiac output (CO), stroke volume (SV), and systemic vascular resistance (SVR).

**Figure 2 jcm-14-02228-f002:**
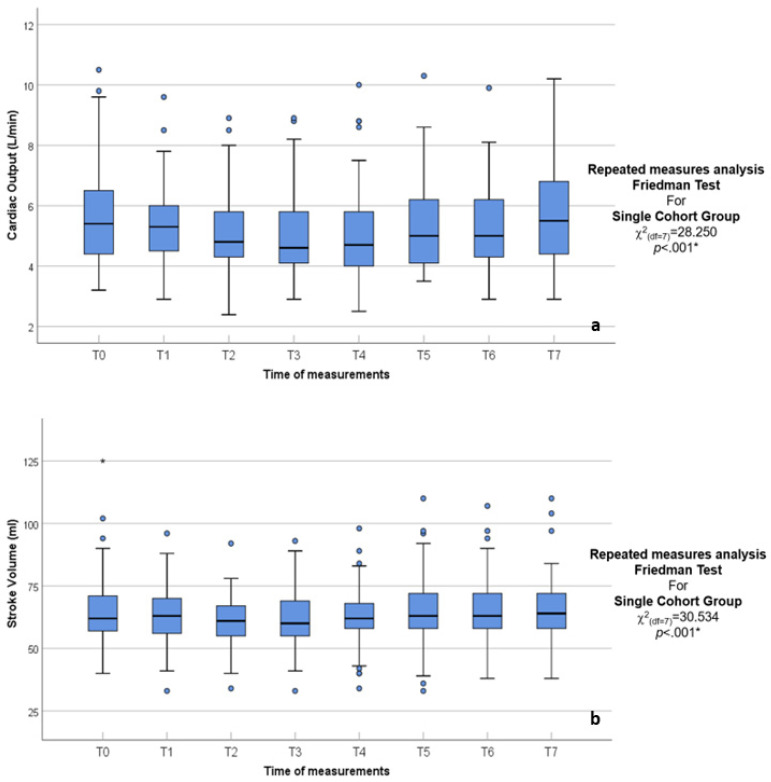
(**a**,**b**) Box and whisker graph of CO L/min and SV mL in the studied group, the thick line in the middle of the box represents the median, the box represents the inter-quartile range (from 25th to 75th percentiles), the whiskers represent the minimum and maximum after excluding outliers (circles), and extremes (asterisks).

**Figure 3 jcm-14-02228-f003:**
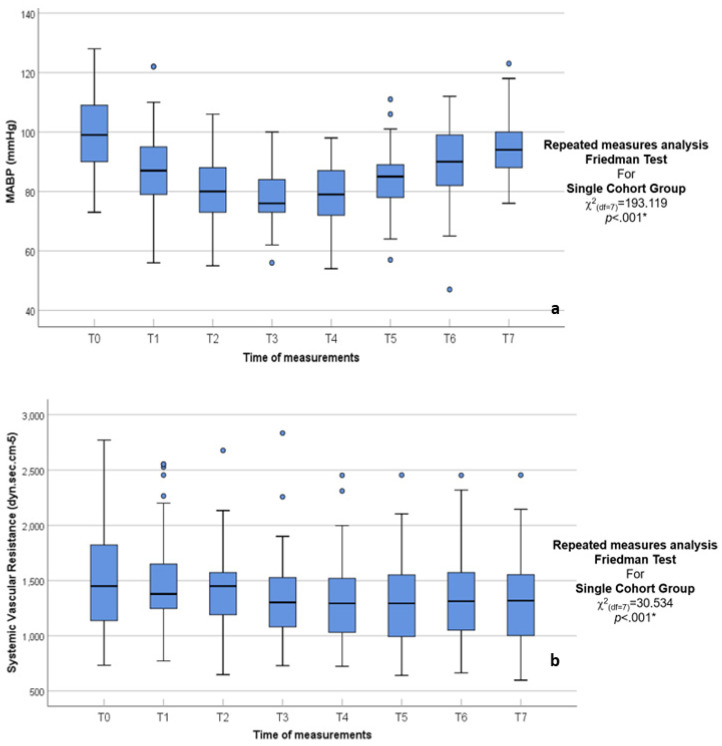
(**a**,**b**) Box and whisker graph of MABP mmHg and SVR in the studied group, respectively, the thick line in the middle of the box represents the median, the box represents the inter-quartile range (from 25th to 75th percentiles), the whiskers represent the minimum and maximum after excluding outliers (circles), and extremes (asterisks).

**Figure 4 jcm-14-02228-f004:**
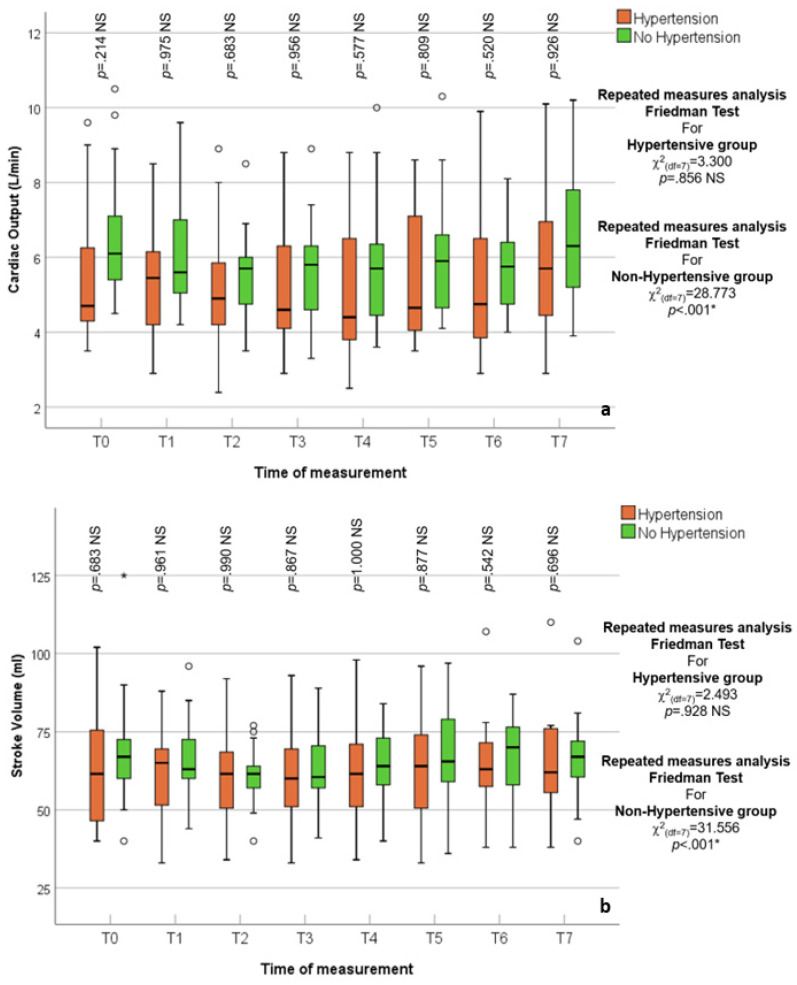
(**a**,**b**) Box and whisker graph of CO L/min and SV mL in the studied groups (Hypertensive vs. Normotensive), respectively. The thick line in the middle of the box represents the median, the box represents the inter-quartile range (from 25th to 75th percentiles), the whiskers represent the minimum and maximum after excluding outliers (circles), and extremes (asterisks).

**Figure 5 jcm-14-02228-f005:**
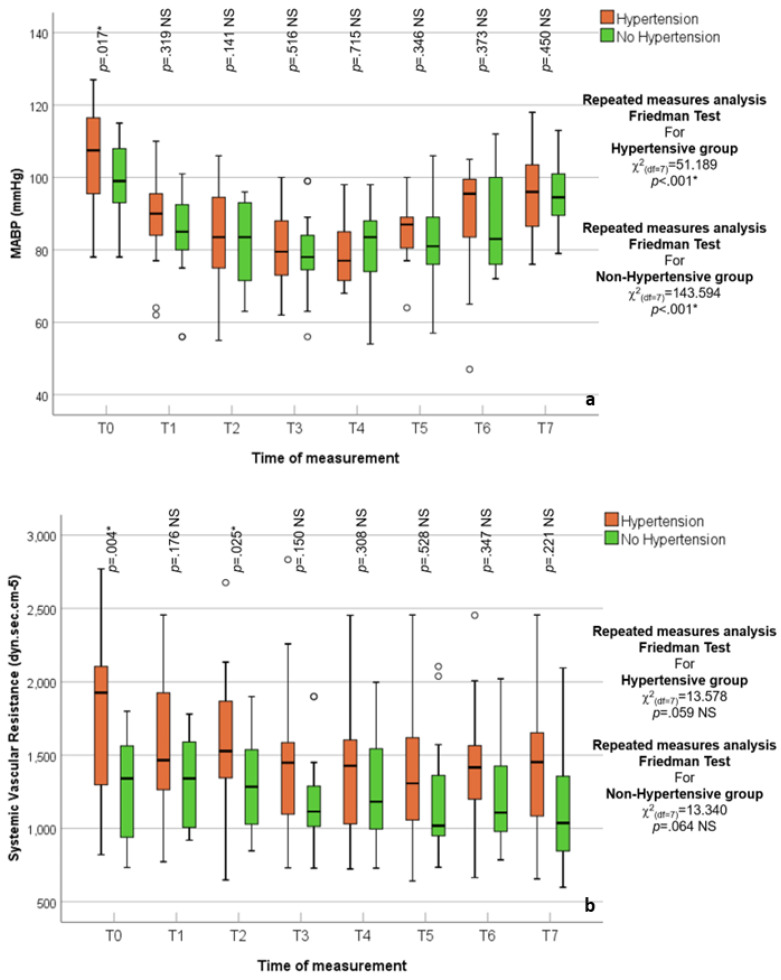
(**a**,**b**) Box and whisker graph of MABP mmHg and SVR in the studied groups (Hypertensive vs. Normotensive), respectively. The thick line in the middle of the box represents the median, the box represents the inter-quartile range (from 25th to 75th percentiles), the whiskers represent the minimum and maximum after excluding outliers (circles), and extremes (asterisks).

**Figure 6 jcm-14-02228-f006:**
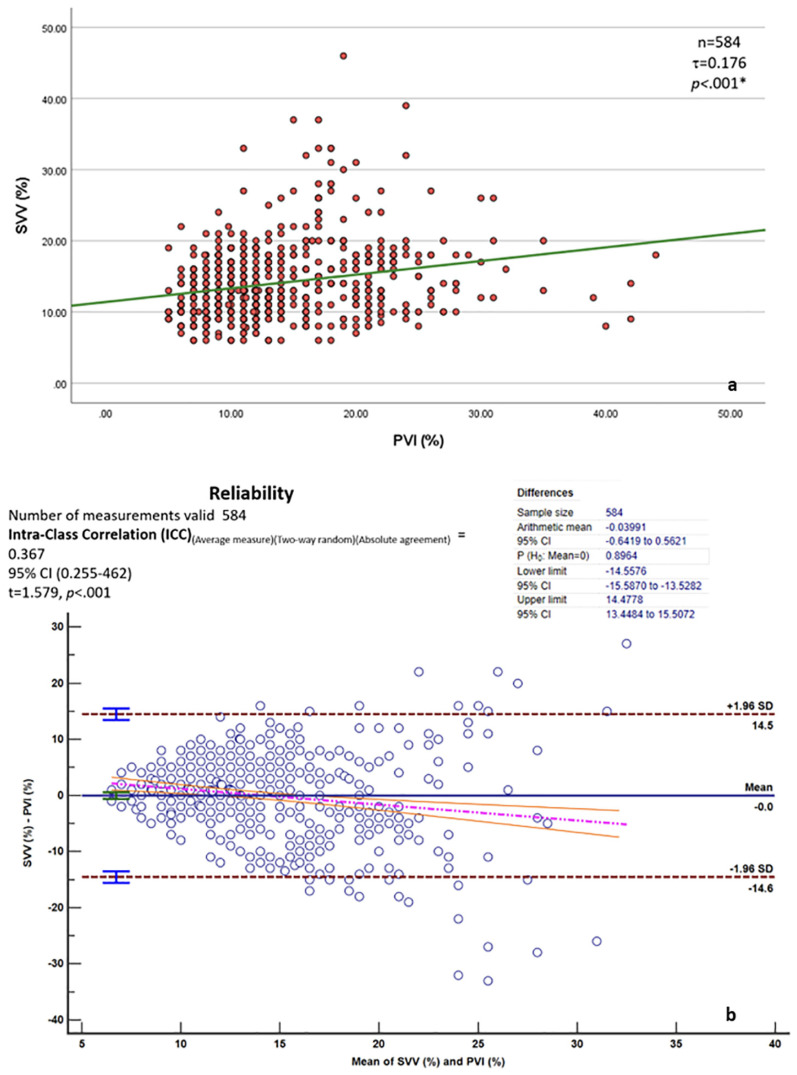
(**a**,**b**) Correlation and agreement between stroke volume variation (SVV. %) of EC (Osypka, Berlin, Germany) and Pleth Variability Index (PVI, %) (Masimo, Irvin, CA, USA). * *p* <0.01 means statistical significance.

**Table 1 jcm-14-02228-t001:** Heart rate (HR), Cardiac output (CO), stroke volume (SV), mean arterial blood pressure (MABP), and systemic vascular resistance (SVR) at each measurement time compared to T0 statistically. T0: Pre-induction; T1: post intubation; T2: post insufflation and positioning; T3: 15 min post insufflation; T4: 30 min post insufflation; T5: end of surgery; T6: 10 min post deflation; T7: post extubation.

	Pre-Induction (T0)	Post-Intubation (T1)	Post-Insufflation and Positioning (T2)	15 min Post-Insufflation (T3)	30 min Post-Insufflation (T4)	End of Surgery (T5)	10 min Post-Deflation (T6)	Post-Extubation (T7)	Test of Significance *p*-Value
Heart Rate (beat/min)									χ^2^_(df = 7)_ = 52.455 *p* < 0.001 *
- Min.–Max.	57.00–120.00	57.00–114.00	58.00–122.00	56.00–100.00	54.00–108.00	50.00–118.00	56.00–120.00	59.00–133.00	
- Mean ± S.D.	88.38 ± 13.39	84.75 ± 14.02	83.900 ± 13.79	80.63 ± 11.29	79.89 ± 10.94	80.53 ± 12.27	82.11 ± 11.89	86.88 ± 14.22	
- Median	89.00	86.00	84.00	81.00	79.00	81.00	82.00	87.00	
- 95% CI of the Median	86.00–94.00	83.00–92.00	79.00–87.00	78.00–84.00	78.00–83.00	80.00–84.00	80.00–87.00	84.00–90.00	
- 25th Percentile–75th Percentile	78.00–99.00	76.00–94.00	75.00–92.00	73.00–88.00	76.00–86.00	74.00–89.00	78.00–88.00	80.00–92.00	
Adjusted *p* value		*p* = 0.202 NS	*p* = 0.553 NS	*p* < 0.001 *	*p* < 0.001 *	*p* < 0.001 *	*p* < 0.001 *	*p* = 1.000 NS	
CO (L/min)									χ^2^_(df = 7)_ = 28.250 *p* < 0.001 *
- Min.–Max.	3.20–10.50	2.90–9.60	2.39–8.90	2.90–8.90	2.50–10.00	3.50–10.30	2.90–9.90	2.90–10.20	
- Mean ± S.D.	5.68 ± 1.58	5.38 ± 1.25	5.08 ± 1.16	5.04 ± 1.34	5.09 ± 1.48	5.33 ± 1.42	5.27 ± 1.32	5.72 ± 1.67	
- Median	5.40	5.30	4.80	4.60	4.70	5.00	5.00	5.50	
- 95% CI of the Median	5.00–5.80	5.10–5.80	4.50–5.40	4.40–5.30	4.30–5.20	4.50–5.60	4.60–5.70	5.00–6.20	
- 25th Percentile–75th Percentile	4.40–6.50	4.50–6.00	4.30–5.80	4.10–5.80	4.00–5.80	4.10–6.20	4.30–6.20	4.40–6.80	
Adjusted *p* value		*p* = 1.000 NS	*p* = 0.012 *	*p* = 0.050 NS	*p* = 0.002 *	*p* = 0.400 NS	*p* = 0.042 *	*p* = 1.000 NS	
SV (mL)									χ^2^_(df = 7)_ = 30.534 *p* < 0.001 *
- Min.–Max.	40.00–125.00	33.00–96.00	34.00–92.00	33.00–93.00	34.00–98.00	33.00–110.00	38.00–107.00	38.00–110.00	
- Mean ± S.D.	64.64 ± 15.00	63.58 ± 11.81	61.00 ± 9.90	61.38 ± 11.58	62.62 ± 11.96	64.59 ± 14.82	65.03 ± 13.08	65.40 ± 13.00	
- Median	62.00	63.00	61.00	60.00	62.00	63.00	63.00	64.00	
- 95% CI of the Median	60.00–67.00	61.00–66.00	59.00–63.00	58.00–63.00	61.00–65.00	60.00–66.00	61.00–68.00	62.00–70.00	
- 25th Percentile–75th Percentile	57.00–71.00	56.00–70.00	55.00–67.00	55.00–69.00	58.00–68.00	58.00–72.00	58.00–72.00	58.00–72.00	
Adjusted *p* value		*p* = 1.000 NS	*p* = 0.056 NS	*p* = 0.047 *	*p* = 0.114 NS	*p* = 1.000 NS	*p* = 1.000 NS	*p* = 1.000 NS	
MABP (mmHg)									χ^2^_(df = 7)_ = 193.119 *p* < 0.001 *
- Min.–Max.	73.00–128.00	56.00–122.00	55.00–106.00	56.00–100.00	54.00–98.00	57.00–111.00	47.00–112.00	76.00–123.00	
- Mean ± S.D.	99.30 ± 12.08	86.33 ± 13.74	80.42 ± 10.62	78.10 ± 9.19	79.07 ± 9.64	84.70 ± 9.72	89.21 ± 11.98	94.49 ± 9.64	
- Median	99.00	87.00	80.00	76.00	79.00	85.00	90.00	94.00	
- 95% CI of the Median	98.00–102.00	85.00–92.00	77.00–83.00	74.00–80.00	76.00–84.00	81.00–88.00	85.00–94.00	92.00–98.00	
- 25th Percentile–75th Percentile	90.00–109.00	79.00–95.00	73.00–88.00	73.00–84.00	72.00–87.00	78.00–89.00	82.00–99.00	88.00–100.00	
Adjusted *p* value		*p*_adjusted_ < 0.001 *	*p*_adjusted_ < 0.001 *	*p*_adjusted_ < 0.001 *	*p*_adjusted_ < 0.001 *	*p*_adjusted_ < 0.001 *	*p* = 0.001 *	*p* = 1.000 NS	
SVR (dyn.s.cm^−5^)									χ^2^_(df = 7)_ = 22.060 *p* = 0.002 *
- Min.–Max.	733.00–2770.00	772.00–2552.00	648.00–2676.00	729.00–2833.00	723.00–2453.00	641.00–2455.00	664.00–2453.00	598.00–2455.00	
- Mean ± S.D.	1498.26 ± 461.83	1472.25 ± 410.74	1425.41 ± 348.13	1337.92 ± 347.90	1330.62 ± 348.24	1300.64 ± 374.51	1351.99 ± 368.01	1322.70 ± 405.56	
- Median	1450.00	1379.00	1450.00	1302.00	1293.00	1293.00	1313.00	1318.00	
- 95% CI of the Median	1284.00–1638.00	1303.00–1561.00	1337.00–1513.00	1267.00–1435.00	1220.00–1419.00	1101.00–1428.00	1196.00–1518.00	1139.00–1451.00	
- 25th Percentile–75th Percentile	1137.00–1823.00	1247.00–1649.00	1191.00–1572.00	1080.00–1527.00	1031.00–1520.00	992.00–1552.00	1051.00–1572.00	1002.00–1553.00	
Adjusted *p* value		*p* = 1.000 NS	*p* = 1.000 NS	*p* = 0.661 NS	*p* = 1.000 NS	*p*_adjusted_ < 0.001 *	*p* = 1.000 NS	*p* = 0.382 NS	

n: Number of patients; Min-Max: Minimum—Maximum; S.D.: Standard Deviation; CI: Confidence interval; NS: Statistically not significant (*p* ≥ 0.05); *: Statistically significant (*p* < 0.05).

**Table 2 jcm-14-02228-t002:** Percentage changes in Heart rate (HR), Cardiac output (CO), stroke volume (SV), mean arterial blood pressure (MABP), and systemic vascular resistance (SVR) and each measurement time compared to T0. T0: Pre-induction; T1: post intubation; T2: post insufflation and positioning; T3: 15 min post insufflation; T4: 30 min post insufflation;T5: end of surgery; T6: 10 min post deflation; T7: post extubation.

		Percentage Change (Versus Pre-Induction)						
	Pre-Induction (T0)	T1 vs. T0	T2 vs. T0	T3 vs. T0	T4 vs. T0	T5 vs. T0	T6 vs. T0	T7 vs. T0
CO (L/min)								
- Min.–Max.	3.20–10.50	−39.22–71.87	−44.90–48.72	−47.14–69.77	−44.44–72.41	−42.86–77.59	−47.78–68.75	−55.56–108.33
- Mean ± S.D.	5.68 ± 1.58	−2.85 ± 18.16	−7.66 ± 19.70	−8.36 ± 22.98	−7.70 ± 24.77	−2.81 ± 26.14	−3.45 ± 25.76	4.03 ± 30.01
- Median	5.40	−2.44	−7.29	−8.62	−11.29	−6.94	−4.55	−1.43
- 95% CI of the Median	5.00–5.80	−6.67–0.00	−15.94–0.00	−15.79–−2.27	−18.97–−5.71	−17.39–0.00	−15.91–0.00	−10.53–4.08
- 25th Percentile–75th Percentile	4.40–6.50	−11.46–4.00	−24.14–4.17	−23.08–2.63	−23.53–0.00	−21.05–7.89	−23.08–7.02	−16.13–26.67
SV (mL)								
- Min.–Max.	40.00–125.00	−40.20–55.00	−38.40–52.50	−36.00–52.50	−36.00–65.85	−34.55–68.29	−40.20–75.61	−39.22–102.50
- Mean ± S.D.	64.64 ± 15.00	−0.08 ± 13.27	−3.55 ± 14.11	−3.12 ± 15.88	−0.86 ± 19.00	1.72 ± 21.28	2.81 ± 19.75	3.67 ± 21.77
- Median	62.00	−1.47	−4.23	−4.76	−3.77	−3.51	−1.59	0.00
- 95% CI of the Median	60.00–67.00	−3.75–1.54	−7.69–0.00	−8.45–0.00	−7.46–0.00	−6.45–1.56	−4.55–5.26	−1.52–4.92
- 25th Percentile–75th Percentile	57.00–71.00	−5.56–4.92	−11.94–5.00	−11.43–4.00	−13.70–7.50	−10.45–7.78	−9.52–14.93	−9.52–10.00
MABP (mmHg)								
- Min.–Max.	73.00–128.00	−48.23–89.43	−49.42–52.22	−49.71–55.24	−61.59–57.03	−58.19–120.32	−56.82–83.68	−63.98–124.21
- Mean ± S.D	99.30 ± 12.08	1.51 ± 20.89	−0.22 ± 24.11	−5.71 ± 25.46	−5.49 ± 27.90	−7.82 ± 30.48	−4.31 ± 29.64	−6.92 ± 31.28
- Median	99.00	−1.31	−2.98	−9.31	−3.09	−14.57	−5.44	−12.32
- 95% CI of the Median	98.00–102.00	−3.29–2.34	−6.03–5.56	−15.39–5.56	−10.00–7.42	−19.04–2.47	−16.28–4.03	−16.37–−1.46
- 25th Percentile–75th Percentile	90.00–109.00	−8.43–10.00	−16.38–17.59	−25.92–11.76	−28.98–11.40	−31.55–8.77	−28.10–14.49	−29.11–7.45
SVR (dyn.s.cm^−5^)								
- Min.–Max.	733.00–2770.00	−40.40–22.00	−51.33–9.09	−44.09–17.95	−46.09–25.64	−42.42–28.21	−60.50–20.51	−32.74–28.13
- Mean ± S.D.	1498.26 ± 461.83	−12.62 ± 12.99	−18.13 ± 12.75	−20.12 ± 14.11	−19.16 ± 14.19	−13.57 ± 14.01	−9.32 ± 13.15	−3.94 ± 11.42
- Median	1450.00	−14.91	−18.63	−23.16	−20.00	−14.44	−9.09	−4.55
- 95% CI of the Median	1284.00–1638.00	−16.67–−9.47	−21.74–−14.29	−24.78–−16.85	−23.00–−15.73	−18.75–−11.22	−14.00–−5.95	−6.78–−1.01
- 25th Percentile–75th Percentile	1137.00–1823.00	−20.95–−2.65	−26.60–−11.21	−29.29–−10.10	−26.26–−10.09	−22.00–−7.14	−17.27–−1.83	−11.00–3.57

## Data Availability

Data are available from the authors upon reasonable request.

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
