# Peer review of "Non-Invasive Cardiac Output Monitoring with Electrical Cardiometry During Laparoscopic Cholecystectomy Surgery, a Cross-Sectional Study"

_jcm, 2025, doi:10.3390/jcm14072228_

Round 1
Reviewer 1 Report
Comments and Suggestions for Authors
Manuscript entitled "Non-Invasive Cardiac Output Monitoring with Electrical Cardiometry During Laparoscopic Cholecystectomy Surgery. A Cross-Sectional Study" by Khaled Ahmed Yassen et al.
This manuscript presents a cross-sectional study evaluating the hemodynamic effects of laparoscopic cholecystectomy (LC) using electrical cardiometry (EC) as a non-invasive monitoring tool. The study provides valuable insights into intraoperative cardiovascular fluctuations associated with increased intra-abdominal pressure (IAP). The methodology is well-structured, and the findings have potential clinical relevance.
Comments
- The introduction appropriately highlights the hemodynamic impact of laparoscopic procedures; however, it should discuss why traditional hemodynamic monitoring methods may be insufficient in laparoscopic surgery. Additionally, it should explain whether electrical cardiometry has been validated against invasive techniques such as pulmonary artery catheters or esophageal Doppler and address potential limitations of this method.
- Were patients with hypertension stratified based on medication use? This could affect systemic vascular resistance (SVR) and cardiac output (CO).
- Were baseline CO and SVR values recorded preoperatively in the outpatient setting? This would help determine if intraoperative changes are surgery-related rather than patient-specific variability.
- Did depth of anesthesia influence hemodynamic parameters? Was BIS monitoring or entropy used?
- Were vasoactive medications used to stabilize hemodynamics, and if so, how was this accounted for?
- Did any patients exhibit paradoxical hypertensive responses to insufflation?
- The manuscript reports that CO and MABP decreased by -47.14% and -61.59% in specific patients; however, it should clarify whether these patients were older, obese, or on beta-blockers, as these factors could affect compensatory mechanisms. Additionally, did any of these patients require intraoperative fluid boluses or vasopressor support?
- Does EC offer advantages over other non-invasive techniques (e.g., pleth variability index, esophageal Doppler)?
- Enhance hemodynamic analysis—include individual patient response patterns, predictive modeling, and longitudinal follow-up.
- Improve clinical applicability—assess whether EC can predict complications or guide fluid/vasopressor therapy.
- Expand discussion of hypertensive subgroup findings—clarify whether preoperative medications influenced responses.
Author Response
Reply to Reviewers
Please add a structured abstract
Thank you. Done and added to the manuscript
Reviewer 1
This manuscript presents a cross-sectional study evaluating the hemodynamic effects of laparoscopic cholecystectomy (LC) using electrical cardiometry (EC) as a non-invasive monitoring tool. The study provides valuable insights into intraoperative cardiovascular fluctuations associated with increased intra-abdominal pressure (IAP). The methodology is well-structured, and the findings have potential clinical relevance.
Comments
- The introduction appropriately highlights the hemodynamic impact of laparoscopic procedures; however, it should discuss why traditional hemodynamic monitoring methods may be insufficient in laparoscopic surgery. Additionally, it should explain whether electrical cardiometry has been validated against invasive techniques such as pulmonary artery catheters or esophageal Doppler and address potential limitations of this method.
- Why traditional hemodynamic monitoring methods may be insufficient in laparoscopic surgery?
Because: Placing invasive catheters for cardiac functions monitoring cannot be justified during routine laparoscopic cholecystectomy unless significant cardiovascular dysfunction is present. This paragraph was mentioned in the introduction.
I agree this point need to be explained.
Another paragraph was added, reworded, and underlined in yellow.
Traditional systemic monitors cannot measure cardiac functions or identify significant dysfunctions, and invasive catheter placement for cardiac function monitoring during routine laparoscopic cholecystectomy is not warranted unless there is significant cardiovascular disease dysfunction.
Heart rate and blood pressure don't provide a full picture of a patient's hemodynamics. The gaps in traditional monitoring can be filled by the parameters that EC offers. Fluid resuscitation and medication therapy can be guided in a targeted, continuous way by EC parameters such CO and SV measurements, which offer additional information on preload, contractility, after load, and delivered oxygen.
- Electric cardiometry validation:
The "gold standard" techniques like thermodilution have been used to validate electrical cardiometry [9]. More was mentioned in details as part of the discussion om page 17 lines 341-361 with related references, but a brief mention was add to the introduction as instructed.
- Potentional limitations are mentioned among discussion in page 17 and 18 lines 363-378 with related references
- Were patients with hypertension stratified based on medication use? This could affect systemic vascular resistance (SVR) and cardiac output (CO).
Controlled hypertensive patients (ASAII) on medication
Added to the text page 11 line 219
Before surgery, all ASA II hypertension patients were under pharmacological control.
Beta blocker and angiotensin receptor blockers arbs
- Were baseline CO and SVR values recorded preoperatively in the outpatient setting? This would help determine if intraoperative changes are surgery-related rather than patient-specific variability.
All baseline measurements were in operating rooms prior to anaesthesia induction. Added to the text page 4 line 139.
- Did depth of anesthesia influence hemodynamic parameters? Was BIS monitoring or entropy used?
Anesthesia depth was monitored throughout the surgery with SEDline from Massimo mentioned in anesthesia technique on page 3 line 115
and inhaled sevoflurane to keep the anesthesia depth between 25 and 50 as per SEDLine anesthesia depth monitor. (Masimo, Irvine, CA, USA)
- Were vasoactive medications used to stabilize hemodynamics, and if so, how was this accounted for?
…catecholamine consumption were all documented. Mentioned on page 4, line 137 it was among the retrieved data, however no patient required any additional support apart from reducing anesthesia depth as mentioned in results section. Page 5 line 170 the following was mentioned
This significant drop in MABP was managed by reducing anesthesia depth until the MABP recovered without the need for any catecholamine support in any patients.
Catecholamine support option was added to the methods section
Any decrease in blood pressure was controlled by lowering the level of anesthesia and maintaining the PVI first, followed by the administration of catecholamine support if required. Page 4 line 129
- Did any patients exhibit paradoxical hypertensive responses to insufflation?
No event was recorded.
Added to results page 5 line 180 and highlighted in yellow.
- The manuscript reports that CO and MABP decreased by -47.14% and -61.59% in specific patients; however, it should clarify whether these patients were older, obese, or on beta-blockers, as these factors could affect compensatory mechanisms. Additionally, did any of these patients require intraoperative fluid boluses or vasopressor support?
Thank you for raising this important point
- There is 8 times of measurements in the study,
- The patients with decrease in their COP more than or equal to 40% (Extreme reduction compared to T0 baseline) during different times of the study are demonstrated in the table below.
- The number of patients with more than 40% decrease in CO, are 1 patient at T1, another one at T2, 4 patients at T3, 3 at T4, 2 at T5 and 3at T6 and 2 at T7.
|
|
Count |
Column N % |
|
|
CO_change_01_category |
No change |
5 |
6.8% |
|
Increased |
27 |
37.0% |
|
|
less than 40% decrease |
40 |
54.8% |
|
|
40% or more decrease |
1 |
1.4% |
|
|
CO_change_02_category |
No change |
6 |
8.2% |
|
Increased |
23 |
31.5% |
|
|
less than 40% decrease |
43 |
58.9% |
|
|
40% or more decrease |
1 |
1.4% |
|
|
CO_change_03_category |
No change |
5 |
6.8% |
|
Increased |
20 |
27.4% |
|
|
less than 40% decrease |
44 |
60.3% |
|
|
40% or more decrease |
4 |
5.5% |
|
|
CO_change_04_category |
No change |
4 |
5.5% |
|
Increased |
18 |
24.7% |
|
|
less than 40% decrease |
48 |
65.8% |
|
|
40% or more decrease |
3 |
4.1% |
|
|
CO_change_05_category |
No change |
6 |
8.2% |
|
Increased |
25 |
34.2% |
|
|
less than 40% decrease |
40 |
54.8% |
|
|
40% or more decrease |
2 |
2.7% |
|
|
CO_change_06_category |
No change |
5 |
6.8% |
|
Increased |
25 |
34.2% |
|
|
less than 40% decrease |
40 |
54.8% |
|
|
40% or more decrease |
3 |
4.1% |
|
|
CO_change_07_category |
No change |
5 |
7.0% |
|
Increased |
29 |
40.8% |
|
|
less than 40% decrease |
35 |
49.3% |
|
|
40% or more decrease |
2 |
2.8% |
|
Here is a full description of the 8 patients whom CO decrease 40% or more throughout the times of operation
All patients with co-existing diseases
- 3 Males and 5 females
- 5 (62.0%) had coexisting disease
- 4 (50.0%) Obese (2 of them had also hypertension)
- 3 (37.5%) hypertensives (2 of them also obese)
- 2 (25.0%) bronchial asthma
- 1 (12.5%) Depression
- 2 (25.0%) Diabetes mellitus
- None required catecholamine support, or extra boluses of fluids, only reduce anesthesia depth was needed.
- The following was added to the manuscript:
Over the course of the procedure, eight patients experienced a 40% or more drop in CO. Three males and five females had co-existing diseases. Five (62.0%) of the eight had a comorbid illness (Three with hypertension, and 4 obese). This significant drop in MABP (Anesthesiologists blinded to EC CO) managed by reducing anesthesia depth until the MABP recovered without the need for any catechola-mine support or extra fluids in any patients.
- Does EC offer advantages over other non-invasive techniques (e.g., pleth variability index, esophageal Doppler)?
PVI do not provide cardiac function and contractility data as the electric cardiometry, however can be used to guide fluid therapy
Esophageal Doppler is a minimal invasive technique in contrast to the noninvasive approach of the elevctric cardiometry the device investigated in current research. In addition inserting the Doppler probe in the esophagus requires more skills than placing the skin electrodes of the EC
In addition, EC offers a couple of advantages over the other non-invasive technologies:
- EC is the only technology that is approved for a wide range of patient ages and weight (from 1 kg weight and day one pre-mature babies, till 250 kg weight) while all other non-invasive technologies are not approved/suitable for all age groups
- Esophageal Doppler and Echo are user dependent from one side (needs trained personnel to guarantee accurate results) and do not offer a full range of parameters that EC provides (while EC is user independent, does not need an expert doctor or technician to operate it)
Esophageal doppler as well can not be used in the pre-operative phase or post-operative care as it requires an unconscious patient) as it needs an unconscious patient)
- PVI can be significantly affected by factors like mechanically ventilated patients, body position, respiratory rate, and the type of anesthesia used, which can limit its accuracy in interpreting fluid responsiveness (which is not the case with EC)
- Medications that affect vascular tone, like vasopressors, can significantly alter the Pleth. waveform and make PVI inaccurate (which is not the case with EC)
- PVI is not reliable in patients with poor peripheral perfusion and severe hypovolemia (while EC works perfectly on those patients)
In Manuscript the following paragraph was added
In contrast to EC, medications that alter vascular tone, such as vasopressors, can drastically affect the pulse oximetry and PVI, routinely used as monitors. Patients with severe hypovolemia and inadequate peripheral perfusion respond best to EC, whereas PVI is unreliable in these cases. Added to Page 15 line 265
- Enhance hemodynamic analysis—include individual patient response patterns, predictive modeling, and longitudinal follow-up.
- If we include patient response patterns the study design will change to now a case series study
- Predictive modelling is unfortunately not in the objectives of the study, but may be in future studies we can include it.
- Longitudinal follow-up is not in the design of the present study, as it is designed as a cross-sectional study
- Improve clinical applicability—assess whether EC can predict complications or guide fluid/vasopressor therapy.
EC can guide fluid management by monitoring the stroke volume variation parameter, however the utilization of this parameter to predict fluid intake was not in the objectives of this current research.
The only analysis was done to correlate the SVV to the PVI as presented in Fig 6 among results.
Figure 6 shows a correlation and agreement between the Stroke Volume Variation (SVV, %) of EC and the Pleth Variability Index (PVI, %, Masimo, Irvin, USA), but the clinical association was poor despite the statistical significance, based on the Rule of Thumb for Interpreting the Size of a Correlation Coefficient by Hinkle DE et al (2003) [26].
- Expand discussion of hypertensive subgroup findings—clarify whether preoperative medications influenced responses.
Preoperative medications particularly beta blockers could have blunted the vascular receptors response, but a future research based on this current study will look into this point in details.
The following paragraph was carefully rewritten and added to the manuscript text:
A new reference was added to support the findings ref 33
The results analysis also identified subgroups of patients with and without hypertension in the current study.
Hemodynamic changes are known to be more pronounced in hypertensive patients than normotensive patients. However, in the current study, the CO trend changes in HTN patients were not substantially affected by inflation or deflation (p=0.8). May be because the HTN patients respond differently due to their altered vascular elasticity. Our inclusion criteria only include ASA I-II patients with controlled hypertension (ASA II). Patients included were on beta-blockers and angiotensin receptor blockers (arbs) therapy. The blunted baroreceptor response, as a result of these medication, could have helped to prevent significant fluctuations in CO and MABP during peritoneal insufflation. Kim et al (2010) came to the same conclusion when they studied hypertensive patients during laparoscopic colectomy and their results indicated that pneumoperitoneum does not lead to clinically negative hemodynamic changes in heart rate, mean arterial pressure or cardiac output of hypertensive patients, who have taken antihypertensive drugs for more than 1 month [33]. Interestingly, the hypertensive patients in our study (all ASA II) were on daily beta-blockers and arbs prior to the scheduled surgery and instructed to take their morning doses. They showed minimal changes in CO and MABP, unlike normotensive patients.
These findings suggest that well-controlled hypertensive patients (ASA I-II) may not be at significantly increased perioperative risk. However, it is essential to emphasize that further studies are needed to determine whether this trend persists in ASA III-IV patients with severe hypertension or with other cardiovascular comorbidities.
No documented hypertensive episodes in current study was reported, it is challenging to assess the impact of these drugs (beta-blockers and arbs) on the patients' hemodynamic response based on the number of patients in current study.
To validate these findings in our research, a larger-scale investigation is advised as confirmed by the post hoc analysis. In support to the above, a recent research by Zhang et al. in 2023 [34], also suggested that hypertensive patients may react differently to high IAP, and they attributed this to their low vascular elasticity and adaptation. The potential for a more personalized intraoperative care in patients with co-existing diseases as hypertension should be encouraged.

Reviewer 2 Report
Comments and Suggestions for Authors
This is an interesting paper. I have several comments.
1. Why is this study justified? Are current monitoring methods insufficient?
2. Is it the authors' opinion that EC should be integrated into routine monitoring?
3. Why was laparoscopic cholecystectomy (LC) chosen as the surgical method? LC is a relatively simple procedure, often requiring less than 30 minutes of operation time. Of course there are difficult cases requiring more than 1 hour, but it is safe to say that the majority of cases are short, compared to other procedures such as hepatectomy or Whipple operation.
4. What is the additional cost of EC? I can see that there can be benefits for patients with cardiac risk or patients undergoing major operations. But the cost-effectiveness should be addressed for implementing this new technique to routine operations like laparoscopic cholecystectomy. A formal analysis would be difficult at this stage, but any information regarding the cost of this procedure would be helpful.
Author Response
Reviewer 2
This is an interesting paper. I have several comments.
Thank you for your comments, which will help to improve this manuscript.
- Why is this study justified? Are current monitoring methods insufficient?
EC detected significant hemodynamic trend changes that conventional monitors could miss.
The elevated IAP during laparoscopic cholecystectomy has an effect on cardiac and systemic hemodynamics.
The primary cause of the inadequacy of the current vital sign monitoring techniques is that they yield delayed results. Thus, for instance, "if a vital sign monitor abruptly displayed a drop in blood pressure," this indicates that the real alterations within the patient's circulation began earlier. A sudden and severe drop in blood pressure occurs only when the patient's body fails to compensate for the various methods, it tried to use to prevent this drop. The precise cause of this reduction in blood pressure can remain unclear, the EC with its parameters can help at an earlier stage to diagnose and mange any deficits..
- Is it the authors' opinion that EC should be integrated into routine monitoring?
EC demonstrated its potential as an additional useful tool in perioperative monitoring. Its non-invasive nature, combined with real-time cardiac and hemodynamic monitoring, makes it an attractive option for ensuring patient safety beside standard monitors, particularly for patients with cardiac conditions.
The reasons for that is that EC provides real-time changes non-invasively and beat by beat to the hemodynamic changes inside the patient, which is of added value to detect any minor changes within the cardiovascular system on time. (specially for fluid management or any needed inotrope support), Routine monitoring shows delayed results after the body fail to compensate the initial hemodynamic changes “which we can be detect on the spot by EC”
Page 15 line 258 the following was adjusted in text:
EC succeeded to offer a real-time, non-invasive way to track cardiac functions, and highlighted individual differences in hemodynamic responses that conventional monitors could have overlook.
EC's beat-by-beat surveillance of the patient's cardiac and systemic hemodynamic changes adds value to conventional monitors by enabling early detection of any emerging abnormalities in the cardiovascular system. EC can detect the hemodynamic changes on spot, while other routine monitoring provides delayed results when the body fails to adjust.
- Why was laparoscopic cholecystectomy (LC) chosen as the surgical method? LC is a relatively simple procedure, often requiring less than 30 minutes of operation time. Of course there are difficult cases requiring more than 1 hour, but it is safe to say that the majority of cases are short, compared to other procedures such as hepatectomy or Whipple operation.
There have been multiple reports of abrupt intraoperative cardiovascular events during a smooth laparoscopic cholecystectomy, as well as documented unfavorable cardiovascular, hormonal, and neuroendocrine changes, brought on by elevated IAP [7, 8]. Traditional systemic monitors cannot measure cardiac functions or identify significant dysfunctions, and invasive catheter placement for cardiac function monitoring during routine laparoscopic cholecystectomy cannot be justified unless there is significant cardiovascular disease dysfunction prior to surgery.
Now it is possible to monitor a continuous, beat-by-beat CO by the Electrical cardiometry (EC) (ICON monitor; Cardiotronics Inc., La Jolla, CA, USA), which is a noninvasive cardiac output monitor and provide more information regarding cardiac contractility and function
- What is the additional cost of EC? I can see that there can be benefits for patients with cardiac risk or patients undergoing major operations. But the cost-effectiveness should be addressed for implementing this new technique to routine operations like laparoscopic cholecystectomy. A formal analysis would be difficult at this stage, but any information regarding the cost of this procedure would be helpful.
In the Kingdom of Saudi Arabia (KSA), a single-use disposable consumable set costs 650 Saudi Riyals per patient, whereas a monitor device costs 90,000 SR. The price is expected to decrease even more when this single-use item is registered with the National Unified Procurement Company (NUPCO), a government public investment fund-owned entity specifically charged with figuring out the prices of healthcare services.
NUPCO has set significantly higher costs for a range of cardiac output devices and supplies. The minimally invasive FLOTRAC sensor, for example, costs 1386.32 Saudi Arabian Riyals and is compatible with the EDWARDS EV1000 device.
The following text was added to the manuscript: Page 17-18 line 393 and highlighted in yellow
The additional cost of cardiac output monitoring can be justified for patients with cardiac risk or patients undergoing major operations, but the cost for implementing this to routine operations like laparoscopic cholecystectomy need to be addressed.
One single-use disposable EC skin electrodes set costs 650 Saudi Arabia Riyals (SAR) for each patient in the Kingdom of Saudi Arabia (173.3 US Dollar), whereas the device monitor costs 90,000 SAR (or Free for every 400 disposable set). The near future registration of the product with the National Unified Procurement Company (NUPCO), an organization owned by the Saudi Government Public Investment Fund (nupco.com/en/) will reduce the costs further. NUPCO as an organization is expressly tasked with determining and reducing the costs of healthcare services throughout Saudi Arabian hospitals.

Reviewer 3 Report
Comments and Suggestions for Authors
The article Non-Invasive Cardiac Output Monitoring with Electrical Cardiometry During Laparoscopic Cholecystectomy Surgery: A Cross-Sectional Study presents information on the non-invasive measurement of intraoperative individual responses in cardiac and systemic hemodynamics, with a focus on cardiac output. Recommendations:
- Keywords are missing.
- The abstract should be structured.
- The introduction is too brief and lacks sufficient general background information.
- The statistical analysis should include demographic data, multinomial regressions and power of study calculations.
- Lines 335–356 should be removed.
- Conclusions should be presented in a separate section.
Author Response
The article Non-Invasive Cardiac Output Monitoring with Electrical Cardiometry During Laparoscopic Cholecystectomy Surgery: A Cross-Sectional Study presents information on the non-invasive measurement of intraoperative individual responses in cardiac and systemic hemodynamics, with a focus on cardiac output. Recommendations:
- Keywords are missing.
Added to the text
Keywords: Hemodynamics; Cardiac Output; Laparoscopic Cholecystectomy; Electric Cardiometry.
- The abstract should be structured.
Done and added to the manuscript
- The introduction is too brief and lacks sufficient general background information.
Introduction improved
Introduction adjusted and more text added to explain why traditional hemodynamic monitoring methods may be insufficient in laparoscopic surgery. Additionally information also included that the electrical cardiometry has been validated against other Cardiac output invasive techniques such as pulmonary artery catheters or esophageal Doppler. The added text highlighted in yellow
- The statistical analysis should include demographic data, multinomial regressions and power of study calculations.
- The demographics were added in results and appeared intext as follows: The demographics of the 73 included patients were 53 (72.6%) females and 20 (27.4 males), with a mean ± age of 42.6 ± 9.8 years, body mass index (BMI) 26.9[98 - 32.14] kg/m², 28.8% had hypertension, and 27.4 % were men. 38.4% were ASA I. Page 5 lines 164
- Multinomial regressions: Statistics Consultant reply
A multinomial logistic regression (or multinomial regression for short) is used when the outcome variable being predicted is nominal and has more than two categories that do not have a given rank or order.
In the present study, we are not predicting (the study is not a predictive study). We do not have such outcome.
- Power of study calculations. Statistic consultant reply
Power analysis can either be done before (a priori or prospective power analysis) or after (post hoc or retrospective power analysis) data are collected. A priori power analysis is conducted prior to the research study, and is typically used in estimating sufficient sample sizes to achieve adequate power.
Sample size calculation before study was done
2.1. Sample Size Calculation presented on page 2 line 88
The minimum sample size was determined based on a previous study which aimed to prospectively examine the effects of pneumopsritoneum on CO and hemodynamics during laparoscopic surgery [11]. Banerjee et al. in 2021 concluded that patients undergoing laparoscopic cholecystectomy experience hemodynamic changes after pneumoperitoneum insufflation and increased IAP. The sample size was calculated to detect the EC as a non-invasive monitor for CO, and for systemic hemodynamics during laparoscopic surgery. At power of study of 80% (b error accepted = 0.20), and difference in proportion of 10%, a minimal required sample size was found to be 57 patients [12].
- Lines 335–356 should be removed.
Done
Removed
- Conclusions should be presented in a separate section.
Conclusion presented as requested in a separate section

Round 2
Reviewer 1 Report
Comments and Suggestions for Authors
The authors have adequately addressed my comments, and the manuscript can be accepted for publication.
Reviewer 3 Report
Comments and Suggestions for Authors
The authors have made the requested modifications.